# Escaping Plato's Cave:
# JAM for Aligning Independently Trained Vision and Language Models

**Lauren Hyoseo Yoon**
Computation and Neural Systems
California Institute of Technology
Pasadena, CA
laurenhyoon@caltech.edu

**Yisong Yue**
Computation and Mathematical Sciences
California Institute of Technology
Pasadena, CA
yyue@caltech.edu

**Been Kim**
Google DeepMind
beenkim@google.com

## Abstract

Independently trained vision and language models inhabit disjoint representational spaces, shaped by their respective modalities, learning objectives, and architectures. The Platonic Representation Hypothesis (PRH) suggests these models may nonetheless converge toward a shared statistical model of reality. This raises a fundamental question: can we move beyond post-hoc detection of such alignment and explicitly optimize for it? We argue this challenge is particularly important for tasks such as **fine-grained contextual distinctions**—where multiple descriptions share global semantics but differ in subtle compositional details. We tackle this setting with the Joint Autoencoder Modulator (**JAM**), which aligns frozen unimodal models by jointly training modality-specific autoencoders with coordinated reconstruction and cross-modal alignment objectives. We systematically evaluate JAM across three design axes: (i) alignment objectives, introducing our multimodal Spread Loss that outperforms classic contrastive methods; (ii) the layer depth at which alignment is most effective; and (iii) the role of foundation model scale in representational convergence. Our findings show that JAM reliably induces alignment (outperforming innately multimodal models and post-hoc alignment baselines with absolute error reduction of up to 10%, and relative error reduction of up to 80%), offering both fundamental insight into the structure of shared semantics and practical guidance for transforming generalist unimodal foundations into specialist multimodal models.

## 1 Introduction

Neural networks trained on different modalities, datasets, and objectives typically inhabit disjoint representational spaces. Yet the Platonic Representation Hypothesis (PRH) [1] suggests that these models—despite having no shared supervision, architecture, or training regime—may nonetheless converge toward a common statistical model of reality. The naming of PRH is inspired by Plato's Allegory of the Cave [2], which describes individuals who observe only shadows of objects cast on the wall of a cave and mistake these projections for the entirety of reality. This hypothesis has been discussed under several philosophical and empirical lenses, including convergent realism in the philosophy of science and the Anna Karenina scenario [3] in representation learning, which suggests that all well-performing models may ultimately resemble each other.

Preprint.

To date, most of the evidence for PRH has been purely observational and coarse-grained. These methods quantify global correlations across feature spaces, but do not provide practical mechanisms for constructing multimodal systems from unimodal ones, nor do they illuminate where alignment fails. We are particularly interested in limitations that arise in *fine-grained contextual settings*, where multiple candidate descriptions may share overall semantics yet differ in specific details. Here, "context" refers to the high-level semantics shared across modalities, while "fine-grained" refers to resolving distinctions within that shared context (e.g., attributes, relations, or localized compositional shifts). For example, distinguishing whether an image contains a dog is a coarse judgment, but deciding between "a brown dog chasing a red ball" and "a brown dog chasing a blue ball" requires contextual sensitivity to subtle compositional cues. See Figure 1 for more examples.

## 1.1 Escaping Plato's Cave (Platonic Alignment)

We study **Platonic Alignment**, a conceptual framework to explicitly align unimodal models trained independently on distinct modalities. Our Joint Autoencoder Modulator (**JAM**, Fig. 3) aligns frozen language and vision representations using coordinated reconstruction and alignment objectives. Reconstruction preserves modality-specific information, while a shared bottleneck enforces a coherent conceptual space capable of aligning both coarse and fine-grained contextual semantics through our newly proposed Spread Loss.

Our contributions are as follows:

- We demonstrate that the existing statistical tests for probing alignment only captures the representational similarity at coarse-level, but not on fine-grained context. Thus, in this fine-grained contextual settings, the representations are still trapped in Plato's Cave.

- We introduce the multimodal **Spread Loss**, which leverages contextual structure to outperform classic contrastive objectives for fine-grained alignment.

- We demonstrate the versatility of our **JAM** framework across a wide range of pretrained backbones. On tasks that require fine-grained visio-linguistic compositional reasoning, we show that **JAM with Spread Loss** consistently provides superior performance over jointly trained multimodal baselines, with an absolute error reduction of up to 10%, and relative error reduction of up to 80%.

- We analyze the impact of pretrained backbone/model's scale and layer depth on alignment performance, providing insights into how these factors interact with alignment supervision.

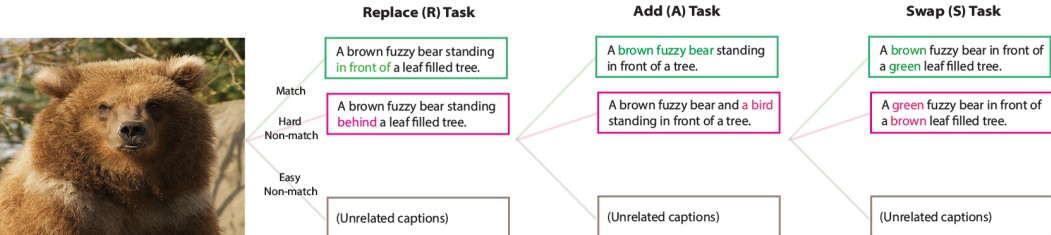

Figure 1: Illustration of fine-grained contextual understanding from the SugarCrepe dataset [4]. Each image is paired with three types of captions: (i) **Match** (true positive) captions that correctly describe the image, (ii) **Easy non-match** captions that are entirely unrelated, and (iii) **Hard non-match** (hard negative) captions that share global semantics with the true caption but diverge in subtle, fine-grained details (e.g., swapping relations, replacing objects, or adding attributes). These controlled perturbations—Replace (R), Swap (S), and Add (A) from the SugarCrepe [4] & Winoground dataset [5] provide a principled test for whether alignment methods capture *contextual fine-grained distinctions* rather than just coarse semantic similarity.

## 1.2 Related Work

Our work is related to two research threads: analyzing correlations between unimodal representations, and aligning different data modalities to construct multimodal models.

**Testing for Alignment.** Prior work in this direction has been largely diagnostic focusing on evaluating alignment between frozen features with broad, context-agnostic datasets (e.g., Wikipedia caption dataset (WIT) [6]): measures such as centered kernel alignment (CKA) [7], variants of CCA (SVCCA [8], projection-weighted CCA [9]), and nearest-neighbor metrics [10, 1]. Other approaches explore probing tasks or zero-shot transfer to assess latent compatibility across modalities [11, 12]. More

recently, model-stitching frameworks have examined whether independently trained sub-networks can be functionally composed to perform new tasks [3], hinting at deeper interoperability between pretrained systems but focusing on vision domain [3, 13].

While informative, these methods are designed to be passive—they measure alignment but do not offer mechanisms to optimize or induce it. As a result, they may conflate superficial correlation with functional compatibility: two embedding spaces may appear statistically aligned yet remain ineffective for fine-grained multimodal tasks. Moreover, although recent state-of-the-art multimodal models (e.g., Gemini [14], GPT-4V [15], LLaMA 3 [16]) demonstrate strong cross-modal performance, they do not explicitly address the alignment dynamics between independently trained unimodal components. In contrast, our work provides a systematic and controlled framework for probing and optimizing cross-modal alignment—offering a potential design for future multimodal systems that build on or unify strong unimodal foundations in specialist settings (which we showcase through fine-grained vision-language compositional tasks).

**Optimizing Alignment of Representations.** A common framing of alignment in multimodal learning refers to the emergence of structurally coherent or comparable latent spaces across modalities, such that semantically related inputs (e.g., images and their captions) map to nearby embeddings. This framing underlies much of the recent progress in large-scale multimodal models such as CLIP [17], ALIGN [18], and BLIP/BLIP-2 [19, 20], DeepSeek [21], which are trained end-to-end using massive paired corpora using explicitly multimodal objectives. Notably, BLIP/BLIP-2 adopt a modular architecture that connects frozen vision encoders and large language models, offering design flexibility. However, the pretrained vision encoders utilized in this method is CLIP, thus inheriting CLIP's multimodal alignment objective from the outset. In contrast, our approach begins with truly unimodal foundations—vision and language models trained independently—and investigates whether alignment can emerge post hoc, without relying on pre-imposed multimodal inductive biases.

## 2 Statistical Tests for Representation Alignment

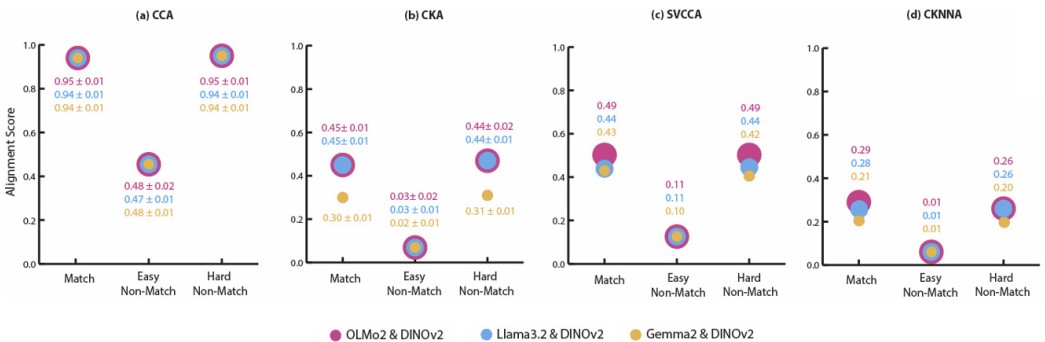

Figure 2: Statistical Metrics for Representation Alignment: Across all metrics and model cases, match pairs consistently show higher alignment scores than easy non-match pairs, supporting the hypothesis that unimodal models encode shared global structure. However, hard non-match pairs exhibit similarly high scores. This indicates that while statistical metrics for representations reveal coarse representational compatibility – consistent with PRH [1] – but are insufficient for diagnosing semantic alignment at a more granular level. (a) Linear and Kernel (Gaussian (i.e., RBF) kernel) PCA to reduce the embeddings dimension to 50; Set CCA dimension as 50. Variation of score values are based on the utilized kernel. (b) Variation of score values are based on the utilized kernel (Linear & RBF). (c) SVD to reduce embeddings dimension to 10; Set CCA dimension as 10. (d) Set top k nearest neighbors = 10 (following the default setting of [1]).

To probe the potential for aligning unimodal models, we applied four alignment metrics—CCA, CKA, SVCCA, and CKNNA—to three types of image–text pairs, illustrated in Fig. 1: **match** (true positive, image–caption pairs), **easy non-match** (unrelated captions), and **hard non-match** (semantically similar captions differing in fine-grained details). The hard negatives are particularly interesting since they share global semantics with the true captions but diverge in localized compositional attributes, allowing us to test whether alignment captures fine-grained context rather than only coarse similarity.

We evaluate embeddings from unimodally pretrained vision and language models (Gemma2 (2B) [22], Llama3.2 (1B) [16], OLMo2 (7B) [23] for language; DINOv2 (ViT-B) [24], ResNet50 [25] for vision), using the SugarCrepe dataset [4], which is explicitly designed for fine-grained vision–language

compositionality. SugarCrepe provides minimal caption perturbations across three transformation families—Replace, Swap, and Add—yielding controlled hard negatives (see Fig. 1). For experiments, we extract CLS-token image embeddings and penultimate-layer text embeddings, then compute alignment scores using the four metrics (See Appendix for metrics formulations).

With the extracted feature representations, let $D$ be the whole data, and $B$ be the batch. We construct $D$ in the nested pair format: $D = \{(v_i, (l_{P_i}, l_{N_i})\}_{i=1}^n$ (see Table 6 for variable descriptions). $C(i)$, the set of similar context text embedding for anchor $i$, is defined as $l_{P_i}$ and its hard negative $l_{N_i}$ (i.e., $C(i) = \{l_{P_i}, l_{N_i}\}$). Accordingly, $\widetilde{C}(i) = L \setminus C(i)$.

## 2.1 Statistical tests detect alignment, but not for fine-grained context

Intuitively, if the Platonic Representation Hypothesis from [1] strongly holds, then the observational alignment metrics should be highest for **match** pairs, second highest for **hard non-match** pairs, and lowest for **easy non-match** pairs. We see in Figure 2 that **easy non-match** pairs indeed do score much lower than **match** pairs across all metrics and models, suggesting that these metrics can detect coarse alignment which is consistent with [1]. However, we also see that **hard non-match** pairs score equally highly as **match** pairs, suggesting that either the models do not share a common representation that can capture fine-grained contextual semantics, or these purely observation alignment metrics are not sufficiently powerful. To move beyond this limitation, we turn to a lightweight post-hoc training approach that explicitly optimizes for semantic coherence, discussed in the next section.

# 3 Our method: Joint Autoencoder Modulator (JAM)

Our approach learns a joint autoencoder across two disjoint pre-trained models, as shown in Figure 3. We first utilize a modality-specific autoencoder for each pre-trained model which distills $d_{input} \rightarrow d_{latent}$, which can be viewed as a form of statistical regularisation reminiscent of PCA [26, 27]: the network is encouraged to find a compact set that preserve essential variance while discarding noise through reconstruction loss (i.e., MSE Loss). The joint training

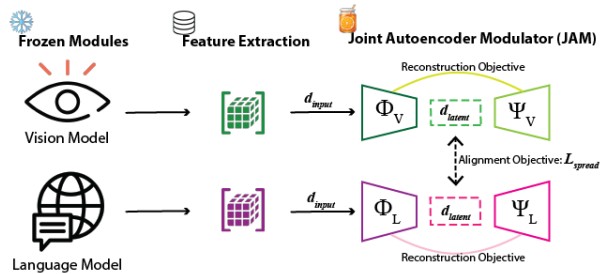

Figure 3: Joint Autoencoder Modulator (JAM) framework.

objective then contrastively pulls the latent vectors from matched image–text pairs together and pushes hard-negative pairs apart.

**Architecture.** The encoder follows a "funnel" layout: a sequence of fully connected layers that progressively reduce the dimensionality of the input embedding, each stage wrapped in Layer-Norm for stability [28], a SwiGLU non-linearity for gated expressiveness [29], and dropout for regularisation. After every dense layer, a lightweight MLP residual block

Table 1: Glossary of variables and symbols

| Symbol | Description |
| --- | --- |
| $\Phi_V, \Psi_V$ | Vision (image) autoencoder (encoder, decoder) |
| $\Phi_L, \Psi_L$ | Language (text) autoencoder (encoder, decoder) |
| $V$ | Set of image embeddings ($|V| = n, v_i \in V, i = \{1, \ldots, n\}$) |
| $L$ | Set of text embedding (i.e., $L_P \cup L_N$; $|L| = 2n$) |
| $L_P$ | Set of positive text embeddings ($|L_P| = n, l_{P_i} \in L_P$) |
| $L_N$ | Set of **hard** negative text embeddings ($|L_N| = n, l_{N_i} \in L_N$) |
| $d_{input}, d_{latent}$ | Input, Latent (i.e., bottleneck) dimension of autoencoder |
| $l_i$ | Element of $L \setminus \{l_{P_i}\}$ with anchor index $i$ |
| $C_L(i)$ | Set of similar context text embeddings |
| $\widetilde{C}_L(i)$ | Set of dissimilar context text embeddings (i.e., $L \setminus C_L(i)$) |

[30] re-injects the intermediate representation back into itself, ensuring gradient flow. In a bottleneck stage, a final linear projection maps the hidden width to the fixed latent size. The decoder mirrors the encoder, walking back through the hidden sizes in reverse. We chose this design to be an effective framework for building compact yet powerful module that fits both high-dimensional text and vision representations (i.e., embeddings). In our experiments, we used 3 hidden layers with dimension size of 512, and bottleneck dimension size as 256.

## 3.1 Loss functions

In addition to the VAE reconstruction objective, our alignment training objective includes three loss functions: the standard Contrastive and Negative Contrastive losses, as well as our novel Spread Loss.

**Contrastive Loss (Con)**  Same as the contrastive loss for training CLIP model [17], we formulate our contrastive loss ($\mathcal{L}_{con}$), in a symmetric fashion, designed to enforce bidirectional consistency between images and texts: Vision-to-Language (VL) as matching each image with its corresponding text, and Language-to-Vision (LV) as matching each text with its corresponding image. We then leverage cross-entropy to maximize the similarity between the correct pairs while minimizing the similarity between incorrect pairs. The cross-entropy loss is given by $\mathcal{L} = -\sum_{i=1}^{N} y_i \log p_i$, where $y_i$ is the ground truth label ($y_i = 1$ for the correct pair and $y_i = 0$ for incorrect pairs), $p_i$ is the predicted probability of the correct pair. Thus, the final contrastive loss is formulated as: $\mathcal{L}_{con} = \frac{1}{2}[\mathcal{L}_{\text{VL}} + \mathcal{L}_{\text{LV}}]$. where image-to-text loss (VL) and text-to-image (LV) loss are the following, respectively:

$$\mathcal{L}_{\text{VL}} = -\frac{1}{N}\sum_{i=1}^{N}\log\frac{\sigma(v_i, l_{P_i})}{\sum_{j=1, j\neq i}^{N}\sigma(v_i, l_{P_j})}, \quad \mathcal{L}_{\text{LV}} = -\frac{1}{N}\sum_{i=1}^{N}\log\frac{\sigma(l_{P_i}, v_i)}{\sum_{j=1, j\neq i}^{N}\sigma(l_{P_i}, v_j)}. \tag{1}$$

Here, we denote $\sigma(x, x') = \exp(\text{sim}(x, x')/\tau)$, where $x, x'$ are the modality-specific embeddings (in this case, text and image embeddings, respectively), and $\tau$ is the temperature parameter that controls the sharpness of the similarity distribution. For all the experiments, we set $\tau = 0.07$, as done in [17].

**Negative Contrastive Loss (NegCon)**  The standard CLIP objective treats all non-matching text descriptions as implicitly negative, but without distinguishing between truly unrelated captions and hard-to-distinguish negative examples (hard negatives)—those that share high semantic similarity with the positive caption but are incorrect due to fine-grained distinctions (e.g., a location or object mismatch). This lack of granularity can lead the model to underutilize structurally informative negative samples. We address this challenge by introducing NegCon loss ($L_{NegCon}$), which allows hard negative texts to also be penalized in the image-to-text direction, while keeping the text-to-image direction unchanged for stability. This loss scheme to incorporate hard negative texts is what has been used in the development of NegCLIP [31].

For the vision-to-language direction, we extend the candidate pool of possible text matches by concatenating both positive and hard negative captions, effectively doubling the number of candidates: The image-to-text loss is then defined as:

$$\mathcal{L}_{\text{VL}} = -\frac{1}{2N}\sum_{i=1}^{N}\log\frac{\sigma(v_i, l_{P_i})}{\sum_{j=1, j\neq i}^{2N}\sigma(v_i, l_j)}. \tag{2}$$

This formulation explicitly anchors the image to its positive caption while contrasting it against a broader and more semantically informative set that includes hard negatives.

For the language-to-vision direction, we retain a standard CLIP-style formulation, utilizing $\mathcal{L}_{\text{LV}}$ utilized in $\mathcal{L}_{con}$. Since hard negative captions have no associated image, we only use positive captions and match them against the corresponding images.

The final NegCon loss is: $\mathcal{L}_{NegCon} = \frac{1}{2}[\mathcal{L}_{\text{VL}} + \mathcal{L}_{\text{LV}}]$. This symmetric structure ensures that both modalities contribute to alignment, while the asymmetry in negative usage avoids destabilizing supervision on text inputs with unknown image counterparts. Hence NegCon provides a minimally modified but effective baseline for studying the role of hard negative supervision.

**Spread Loss**  Recall that our objective is to develop a loss function that more faithfully captures the structure of semantic similarity and fine-grained contrast in multimodal representation learning. Specifically, we aim to *align image and text embeddings in a way that accounts for both shared contextual meaning and localized distinctions (fine-grain)*. Standard contrastive losses treat all non-matching text as equally negative, which neglects *semantic proximity of hard negatives*—text descriptions that differ only in the fine detail.

To address this, we introduce $\mathcal{L}_{spread}$, a contrastive objective that incorporates a notion of context similarity and fine-grained differentiation. As shown in Figure 4, the core idea is to pull together all semantically similar captions, including the ground-truth caption

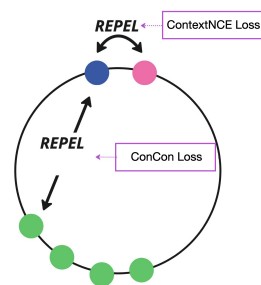

Figure 4: Illustration of the Spread Loss: Blue and Pink circle correspond to similar context group; Green circles are representations outside of similar context. Figure inspired by [32].

and selected hard negatives, and then selectively push apart fine-grained distinctions within that similar group. This two-stage formulation allows us to preserve shared contextual alignment while enforcing discriminative power at the instance level. The idea of spread loss has been proposed in

the domain of visual representation learning [32, 33], which is a variant of Supervised Contrastive (SupCon) loss [34], but tailored to tackling class collapse. We take inspiration from that work and apply this idea to the multi-modal alignment problem.[1] Our $\mathcal{L}_{spread}$ is formulated as:

$$\mathcal{L}_{spread} = \frac{1}{2}\left[(1-\alpha)\mathcal{L}_{ConCon} + \alpha\mathcal{L}_{contextNCE} + \mathcal{L}_{LV}\right], \tag{3}$$

where $\mathcal{L}_{LV}$ is from Equation 1, preserving bidirectional learning throughout joint-AE training. $\mathcal{L}_{ConCon}$ and $\mathcal{L}_{contextNCE}$ are described below.

$\mathcal{L}_{ConCon}$ : ConCon (i.e., Context Contrastive) lumps the positive & hard negative (which share contextual attributes) case as the similar context set and tries to push this set away from all other text embeddings. The objective of this loss is to introduce the global learning scheme where we focus on learning what is easily differentiable. Formally, we construct the loss as follows:

$$\mathcal{L}_{ConCon}(i, B) = -\frac{1}{|C_L(i)|} \sum_{c_l \in C_L(i)} \log \frac{\sigma(v_i, c_l)}{\sigma(v_i, c_l) + \sum_{\widetilde{c_l} \in \widetilde{C_L}(i)} \sigma(v_i, \widetilde{c_l})}.$$

$\mathcal{L}_{contextNCE}$ : We introduce this second term for local learning scheme. We perform an InfoNCE-style objective, explicitly highlighting the true positive as the only "correct" text. The negative text embedding, though labeled as 'similar context', is still in the denominator and is therefore treated as incorrect text.

$$\mathcal{L}_{contextNCE}(i, B) = -\log \frac{\sigma(v_i, l_{P_i})}{\sum_{c_l \in C_L(i)} \sigma(v_i, c_l)}.$$

Thus, $\alpha$ in Equation 3 controls the trade-off between context-level alignment and intra-context contrast. While $\mathcal{L}_{ConCon}$ allows similar context to be close, $\mathcal{L}_{contextNCE}$ prevents representation collapse in similar context embeddings. Thus, $\mathcal{L}_{spread}$ enables learning representations that are not only globally aligned across modalities but also sensitive to subtle mismatches in local content. In our experiments, we use $\alpha = 0.5$ to balance the two components.

## 4 Experiments

We adopt the same data set-up as in the statistical tests using Sugarcrepe [4] and Winoground [5] (see Fig. 1). Winoground is designed to test visuo-linguistic compositionality focused on the Swap (S) task style of Sugarcrepe as described in Fig. 1. We split the data into 70-15-15 train/validation/test.

To test on our framework's versatility across different pretrained backbones, we extracted text embeddings and image embeddings from wider set of models: for language models, Gemma2 (2B,9B) [22], Llama3.2 (1B,3B) [16], OLMo2 (7B,13B) [23], and for vision models, DINOv2 [24], MAE [35], and Swav [36] for self-supervised learning (SSL) objective and ResNet50 [25], Swin [37], and ViT [38] for supervised (Sup) objective. (In Table 2, we show results for DINOv2 and RestNet50 for vision backbones in Sugarcrepe & Winoground data setting. For full results (all possible pretrained backbones configurations), refer to Appendix Table 14.)

For each task, we train our Joint Autoencoder Modulator (JAM) with Spread loss for 100 epochs with a batch size of 32, using data seeds 5, 42, and 55. Both autoencoders are optimized jointly using AdamW [39] with gradient clipping (1.0) and a cosine annealing scheduler. The reconstruction loss is weighted by a linearly decaying factor $\lambda(t)$, decreasing from 1.0 to 0.1 over training epochs to gradually emphasize the alignment objective. Every five epochs, we compute image-to-text Recall@1 on the validation set, applying early stopping if no improvement is observed for five consecutive validations. We evaluate on two retrieval settings: (1) binary choice between the positive and its hard negative (standard in fine-grained evaluation [4, 31]), and (2) 5-way choice including three additional distractors. All reported results are averaged across three seed runs.

To contextualize our approach—designed to escape the "Platonic cave" of unimodal models through post-hoc alignment—we compare against CLIP [17], a model natively trained for multimodal representation learning. For fairness, we also fine-tuned CLIP on the same train/val/test splits using the fine-tuning method proposed in [31], although we find that fine-tuning often causes CLIP to overfit. To further motivate our JAM architecture with spread loss, we experimented on simple projection-based methods (linear/non-linear) with our Spread loss as well as JAM with spread loss without reconstruction component (spread w/o reconst.) as ablation experiment (Table 3).

---

[1]We provide a parallel comparison between $\mathcal{L}_{spread}$ of multimodal representation space (i.e., ours) and $\mathcal{L}_{spread}$ of visual representation domain [32, 33] in the Appendix.

| Language Backbone | Vision Backbone | Alignment Method | Sugarcrepe Tasks | | | Winoground Task |
| | | | Replace (R) | Add (A) | Swap (S) | |
|---|---|---|---|---|---|---|
| Gemma2 (2B) | DINOv2 (ViT-B; 86M) | Con | 65.14 | 58.34 | 57.36 | 46.30 |
| | | NegCon | 86.32 | 96.85 | 74.5 | 45 |
| | | Spread | 88.01 | 95.74 | 80.2 | 58.75 |
| Gemma2 (2B) | ResNet50 | Con | 74.23 | 63.02 | 63.75 | 41.30 |
| | | NegCon | 87.28 | 94.80 | 72.27 | 41.30 |
| | | Spread | 87.60 | 94.81 | **82.17** | 58.13 |
| Llama3.2 (1B) | DINOv2 (ViT-B; 86M) | Con | 66.97 | 60.99 | 66.59 | 50 |
| | | NegCon | 83.43 | 94.25 | 68.26 | 48.80 |
| | | Spread | 87.21 | 97.83 | 79.05 | 57.5 |
| Llama3.2 (1B) | ResNet50 | Con | 67.36 | 67.94 | 71.11 | 42.50 |
| | | NegCon | 85.74 | 94.10 | 71.69 | 46.30 |
| | | Spread | 87.17 | 96.85 | **82.17** | **61.30** |
| OLMo2 (7B) | DINOv2 (ViT-B; 86M) | Con | 68.48 | 65.17 | 66.88 | 43.20 |
| | | NegCon | 85.30 | 96.67 | 71.69 | 47.50 |
| | | Spread | **88.32** | 97.41 | 77.62 | 55 |
| OLMo2 (7B) | ResNet50 | Con | 68.01 | 64.35 | 73.38 | 42.50 |
| | | NegCon | 89.69 | 90.12 | 71.38 | 47.50 |
| | | Spread | 86.90 | **98.44** | 80.46 | 57.50 |
| Pretrained CLIP (ViT-B-32/OpenAI) [40] | | | 81.05 | 77.58 | 64.69 | 59.88 |
| Pretrained CLIP (ViT-B-32/LAION-400m) [40, 41] | | | 80.90 | 79.64 | 67.30 | 57.38 |
| Pretrained CLIP (ViT-B-32/SigLIP) [42] | | | 85.01 | 86.56 | 70.97 | 60.75 |
| Finetuned CLIP (ViT-B-32) | | | 74.62 | 92.69 | 67.21 | 58.33 |

Table 2: Image-to-Text Retrieval Results of Joint Autoencoder Modulator (JAM) for Vision-Language Compositionality. We report Recall@1 scores (binary) across three compositional tasks in Sugarcrepe data (Replace, Add, and Swap) and also in Winoground data (which only has Swap task for vision-language reasoning aspect), using models trained with different alignment methods in JAM framework. JAM with Spread loss consistently outperforms contrastive baselines across all tasks and backbones. Moreover, it matches or surpasses several strong pretrained and finetuned CLIP variants, highlighting the effectiveness of structured alignment over independently pretrained representations for fine-grained vision–language reasoning.

| Post-hoc Alignment Method | FLOPs(G) | Sugarcrepe Data Tasks | | |
| | | Replace (R) | Add (A) | Swap (S) |
|---|---|---|---|---|
| Linear Proj. with Spread | 0.06 | 85.95 | 91.94 | 63.33 |
| NonLinear Proj. with Spread | 0.12 | 86.99 | 92.22 | 68.57 |
| JAM with Spread *w/o reconst.* | 3.70 | 83.11 | 92.53 | 74.50 |
| **JAM with Spread** | 3.70 | 88.01 | 95.74 | 80.2 |
| Pretrained CLIP (ViT-B-32/SigLIP) | N/A | 85.01 | 86.56 | 70.97 |
| Finetuned CLIP (ViT-B-32) | 10.48 | 74.62 | 92.69 | 67.21 |

Table 3: Image-to-Text Retrieval Results in Vision-Language Compositionality with Different Post-hoc Alignment architectures (Linear Projection, Nonlinear Projection, Joint Autoencoder Modulator (JAM)) with Spread Loss and ablation result for removing the reconstruction component in Spread Loss. Same evaluation scheme as shown in Table 2. We show results using Gemma2 (2B) and DINOv2 (ViT-B) as language and vision backbone. We also provide FLOPs(G) across alignment methods (including finetuning CLIP) to show that JAM with Spread framework shows the best performance but requires less FLOPs compared to finetuning.

## 4.1 Does JAM enable escaping Plato's Cave?

Table 2 shows are main benchmark results for the Sugarcrepe & Winoground Tasks with binary Recall@1 setting. We see that our JAM method with Spread loss consistently outperforms both the Con and NegCon variants as well as pretrained/finetuned CLIP across tasks, with up to 10% absolute error reduction and 80% relative error reduction versus the best CLIP baseline. We also see that the other losses Con and NegCon can often perform worse than the CLIP baselines. Additional results are found in Table 14 in the Appendix.

We note that our JAM method achieves competitive performance with a lightweight, post-hoc alignment approach, which avoids the need for massive paired datasets and the complexity associated with pretraining large-scale multimodal models like CLIP. This makes our method more robust in low-resource or specialized data domains, where fine-tuning a model like CLIP can lead to overfitting (i.e., losing CLIP's broad priors) and a "representation collapse" [43], as we also observe in Table 2 that fine-tuning CLIP often led to lower performance.

We conducted additional baseline experiments using linear and nonlinear projections over frozen embeddings. Note that for projection-based baselines, the reconstruction component of our Spread loss is inherently absent. As Table 3 shows, our method (JAM+Spread) consistently outperformed projection baselines: while projections achieved lower FLOPs, they suffered from significantly lower accuracy and unstable learning due to the lack of reconstruction. Furthermore, in ablation experiments, removing the reconstruction term led to rapid overfitting, with validation loss rising sharply and models requiring early stopping to avoid collapse. This highlights the critical role of reconstruction as a regularizer, preserving generalizable structure across modalities beyond alignment.

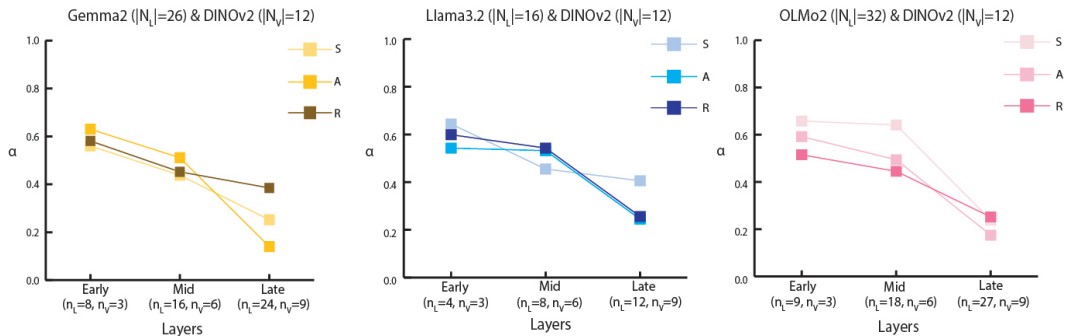

Figure 5: $\alpha$ supervision with respect to the extracted embeddings layers to achieve the best retrieval accuracy for each visuo-linguistic tasks in Sugarcrepe (Replace (R), Add (A), Swap (S) tasks). $|N_L|, |N_V|$ refer to the total layers of each pretrained language, and vision model. $n_L, n_V$ refer to the layer-depth used for Early, Mid, Late experiments, respectively.

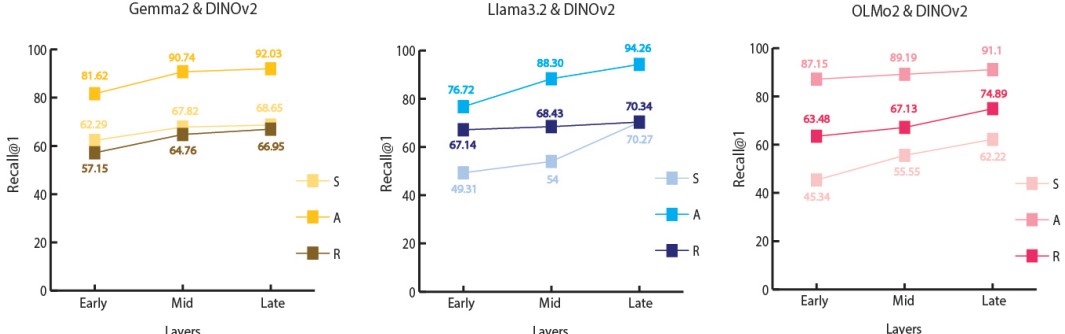

Figure 6: Image-to-Text Retrieval Recall@1 achieved through the $\alpha$ in Figure 5. Same layer-depth configuration as Figure 5. Despite the decreasing need of context-aware supervision through $\alpha$ in $\mathcal{L}_{spread}$, performance increases as layers progress.

## 4.2 What kind of supervision is useful for different layer depth?: Layer-Wise Probing via Curriculum Learning

The previous setting in Table 2 uses a fixed $\alpha = 0.5$, allowing our jointly trained text and image autoencoders to equally emphasize both components of the $\mathcal{L}_{spread}$ — ConCon for broad semantic alignment and ContextNCE for fine-grained contrast. We now investigate how performance changes as we vary this trade-off. In particular, the optimal degree of contextual separation may not be uniform across all settings — in particular, it may vary depending on the stage of representation (i.e., intermediate layers of pretrained backbones) used from each modality.

To investigate this, we move beyond a fixed-$\alpha$ formulation and adopt curriculum learning [44] as a probing framework (Fig. 5, 6). Curriculum learning [44] is a training paradigm inspired by human learning, where models are first exposed to easier concepts or objectives and gradually introduced to harder or more nuanced distinctions. The motivation is to guide optimization along a smoother path: by focusing early training on easier, more generalizable patterns, the model establishes a robust representational foundation before facing fine-grained or ambiguous cases. We use curriculum learning as a diagnostic framework to probe the nature of supervision required for aligning multimodal embeddings.

**On the inverse relationship between $\alpha$ and separability.** By applying our curriculum across embeddings from early, mid, and late layers of vision and language models, we find that earlier layers consistently require higher $\alpha$ for optimal alignment—the point of peak retrieval accuracy (see Figures 5 & 6). This result mirrors those in interpretability studies (e.g., TCAV [45], Network Dissection [46]) showing that lower layers encode primitive or entangled features, demanding stronger supervision to isolate meaningful concepts. Here, larger $\alpha$ provides the finer discriminative signal needed to disentangle and align these early representations. Later layers, already more abstract and task-aligned, align effectively with weaker supervision. Retrieval performance also improves steadily from early to late layers, indicating that deeper layers both require less contrastive pressure and yield more directly alignable representations. Thus, the curriculum offers a controlled lens into the supervision needed across representational depth, revealing how independently trained vision and language models organize information.

### 4.3 Does JAM performance scale with backbone models' scale?

We finally test whether larger pretrained language backbones improve alignment in JAM, using Gemma2 (9B), LLaMA 3.2 (3B), and OLMo2 (13B) with a fixed vision encoder (DINOv2 ViT-L). The aim is to assess how representational capacity affects fine-grained multimodal alignment under different contrastive objectives.

| Language Backbone | Vision Backbone | Alignment Method | Sugarcrepe Data Tasks | | |
|---|---|---|---|---|---|
| | | | Replace (R) | Add (A) | Swap (S) |
| Gemma2 (9B) | DINOv2 (ViT-L; 300M) | Con | 68.77 | 59.31 | 60.92 |
| | | NegCon | 87.52 | 98.89 | 84.44 |
| | | Spread | 89.66 | 98.89 | 84.44 |
| Llama3.2 (3B) | DINOv2 (ViT-L) | Con | 65.24 | 62.38 | 73.12 |
| | | NegCon | 84.54 | 92.61 | 75.93 |
| | | Spread | 89.43 | 95.74 | 82.44 |
| OLMo2 (13B) | DINOv2 (ViT-L) | Con | 65.86 | 59.21 | 62.28 |
| | | NegCon | 89.36 | 93.7 | 78.89 |
| | | Spread | 90.53 | 97.96 | 84.71 |

Table 4: Model Scaling Experiment. We report image-to-text Recall@1 scores across three visuo-linguistic reasoning tasks with Sugarcrepe using language & vision backbones of increased scale—Gemma2 (9B), LLaMA 3.2 (3B), and OLMo2 (13B) for language, and DINOv2 (ViT-L) for vision—within our JAM framework. Despite varying parameter counts, we observe no consistent trend of performance improvement with model size. Spread loss performs robustly across all scales, suggesting that for fine-grained, low-data tasks, representational alignment depends more on objective design than on model scale.

As shown in Table 4, scaling language models does not consistently improve performance. Spread loss remains robust, but gains from larger models are minimal. This plateau likely reflects the small training regime (200–1000 image–text pairs), which limits the gradient signal needed for large models to leverage their capacity. Moreover, the fine-grained nature of spatial reasoning tasks emphasizes precise local alignment (e.g., prepositions, swapped attributes), reducing the advantage of broad generalist knowledge from web-scale pretraining. These results highlight the effectiveness of task-specific Spread loss in capturing nuanced multimodal contrasts.

## 5 Conclusion / Future Directions

Through this framework, we provide empirical evidence and insight into the nature of representational convergence. We show that, despite originating from disjoint modalities and being trained independently, unimodal representations can be aligned through post hoc joint autoencoding — revealing a Platonic representation that supports cross-modal coherence. We propose a practical model training recipe: the Joint Autoencoder Modulator (JAM), a Pareto-efficient framework for building specialist multimodal models on top of frozen unimodal foundations. Our findings show that post-hoc alignment via lightweight adaptation and structured supervision (i.e., $\mathcal{L}_{spread}$) can rival or even outperform generalist architectures in specialist settings. Looking forward, we see JAM as a flexible testing ground for modality alignment under varying supervision regimes.

There are many directions for future work, including extending to domains that demand either highly specific reasoning (e.g., legal, medical, scientific), scaling to larger datasets and models, as well as exploring more sophisticated alignment approaches beyond our Spread loss. At larger scales, it becomes interesting to also compare accuracy-efficiency tradeoffs versus other approaches such as adapter-based and joint-training approaches. Additionally, studying these questions for more than two modalities simultaneously can be interesting, such as for time series [47, 48, 49, 50].

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

# A  Impact Statement

**Platonic Alignment: Escaping the Plato's Cave.**

Platonic Representation Hypothesis [1] suggests that vision and language models, though trained on disjoint objectives and data, encode latent geometries that are mutually compatible. Our work turns this philosophical and observational claim into practice: we *escape Plato's Cave* by *surfacing shared structure from representations originally confined to separate unimodal worlds*, while keeping every pretrained unimodal backbones completely frozen.

**Joint Autoencoder Modulator (JAM).**

We propose JAM, a post-hoc adaptor for platonic alignment that:

- *Preserves specialization* – Modality-specific autoencoders reconstruct each latent space, guarding task-relevant features.
- *Enforces cross-modal coherence* – Our novel spread loss couples the JAM framework, amplifying the latent commonality predicted by the Platonic hypothesis.
- *Light-weight framework* – no end-to-end multimodal fine-tuning (nor pre-training); refer to D.2 that shows the lightweight structure of JAM.

**Contributions.**

- *Platonic Alignment through Specialist Task Settings*: We test the platonic alignment through both statistical and model-training framework in specialist task settings which require more fine-grained contextual understanding of the data.
- *Revisiting multimodal training pipeline strategy*: Our results show that aligning the frozen pretrained backbones can rival with inherently multimodal models, charting a potential alternative strategy for creating multimodal models for specialist task frameworks.

# B  Statistical Metrics to Measure Platonic Alignment

## B.1  Thematic analysis of statistical representation alignment tests

We focus on the following four statistical metrics for testing Platonic alignment: Canonical Correlation Analysis (CCA [51]) with linear and kernel (rbf kernel) PCA, Singular Value CCA (SVCCA [8]), Centered Kernel Alignment (CKA [7]), and Centered Kernel Nearest Neighbors Alignment (CKNNA [10, 1]).

The following are the thematic analyses that demonstrate our metrics selection strategy:

- **Unsupervised and second-order:** All four metrics are *unsupervised* and based on second-order statistics (e.g., inner products or covariances), making them scale-invariant and naturally comparable across modalities.
- **Symmetricity:** The symmetry property indicates that the metric treats the data points interchangeably, meaning $d(x, y) = d(y, x)$.
- **Data-Driven (D) vs. Canonical (C):**
  - **Data-Driven** metrics (CCA, SVCCA) learn linear projections from the data that actively mold one representation space to align with another. They measure the *best-case alignment* achievable through adaptation.
  - **Canonical** metrics (CKA, CKNNA) keep the input representations fixed and evaluate the *existing alignment structure* as is, without transformation. They quantify inherent similarity without retraining or reprojecting.
- **Global (G) vs. Local (L):**
  - **Global** metrics (CCA, SVCCA, CKA) aggregate information across *all sample pairs*, capturing holistic alignment patterns between entire distributions.
  - **Local** metrics (CKNNA) focus on *localized structure* by comparing only mutual $k$-nearest neighbors, revealing fine-grained neighborhood-level similarity that may be obscured globally.

Table 5: Comparative analysis of statistical alignment metrics. All metrics are unsupervised and based on second-order statistics.

| Metric | Property | | | | Description |
|--------|-----------|---|---|---|-------------|
| | Symmetric | Data-Driven(D) vs. Canonical(C) | Global(G) vs. Local(L) | Batchable | |
| CCA | ✓ | D | G | ✓ | Learns projections to maximize cross-view correlation. Sensitive to high-dimensional data structure, hence apply linear or kernel PCA before applying CCA. |
| SVCCA | ✓ | D | G | ✓ | Applies SVD to smooth and compress representations before applying CCA. |
| CKA | ✓ | C | G | ✓ | Compares original representations using kernel alignment, invariant to rotation and scale. |
| CKNNA | ✓ | C | L | ✓ | A local variant of CKA restricted to shared $k$-nearest neighbors; reveals localized alignment structure. |

- **Batchable:** A computational property that makes it feasible to compute in a reasonable time frame.

## B.2 Metrics Formulation

Recall our glossary:

Table 6: Glossary of variables and symbols

| Symbol | Description |
|--------|-------------|
| $\Phi_V, \Psi_V$ | Vision (image) autoencoder (encoder, decoder) |
| $\Phi_L, \Psi_L$ | Language (text) autoencoder (encoder, decoder) |
| $V$ | Set of image embeddings ($|V| = n, v_i \in V, i = \{1, \ldots, n\}$) |
| $L$ | Set of text embedding (i.e., $L_P \cup L_N$; $|L| = 2n$) |
| $L_P$ | Set of positive text embeddings ($|L_P| = n, l_{P_i} \in L_P$) |
| $L_N$ | Set of **hard** negative text embeddings ($|L_N| = n, l_{N_i} \in L_N$) |
| $d_{input}, d_{latent}$ | Input, Latent (i.e., bottleneck) dimension of autoencoder |
| $l_i$ | Element of $L \setminus \{l_{P_i}\}$ with anchor index $i$ |
| $C_L(i)$ | Set of similar context text embeddings |
| $\widetilde{C_L}(i)$ | Set of dissimilar context text embeddings (i.e., $L \setminus C_L(i)$) |

**Kernel preliminaries** Let $\{(\mathbf{x}_i, \mathbf{y}_i)\}_{i=1}^n$ be paired samples: in our case, let $x_i$ be the images and $y_i$ be the corresponding texts. A representation function $f_V : \mathcal{X} \to V \in \mathbb{R}^d$ maps inputs to feature vectors $v_i = f_V(\mathbf{x}_i)$ in an RKHS (Reproducing Kernel Hilbert Space) $\mathcal{H}$ [52] equipped with the inner-product kernel $K_{ij} = \kappa(v_i, v_j) = \langle v_i, v_j \rangle$. Analogously $f_L$ maps $\mathbf{y}_i$ to $l_i$ with kernel $L_{ij} = \langle l_i, l_j \rangle$. Center both kernels to remove mean effects via the centring matrix $H = I - \frac{1}{n}\mathbf{1}\mathbf{1}^\top$:

$$\bar{K} = HKH, \qquad \bar{L} = HLH.$$

### B.2.1 Centered-Kernel Alignment (CKA)

CKA [7] normalises the *Hilbert–Schmidt Independence Criterion* (HSIC) [53] between the centred kernels:

$$\text{HSIC}(K, L) = \frac{1}{(n-1)^2} \text{tr}(\bar{K}\bar{L}), \tag{4}$$

$$\text{CKA}(K, L) = \frac{\text{HSIC}(K, L)}{\sqrt{\text{HSIC}(K, K)\ \text{HSIC}(L, L)}}. \tag{5}$$

Expanding the trace in Equation 4 shows that every pair $(i, j)$ contributes a product of centered similarities:

$$\text{tr}(\bar{K}\bar{L}) = \sum_{i=1}^n \sum_{j=1}^n \big(\langle v_i, v_j \rangle - \mathbb{E}_\ell \langle v_i, v_\ell \rangle\big)\big(\langle l_i, l_j \rangle - \mathbb{E}_\ell \langle l_i, l_\ell \rangle\big). \tag{6}$$

Because the denominator applies the same operation to each modality, CKA is scale-invariant and bounded in $[0, 1]$; it therefore has advantage in measuring *global* geometric correspondence. However, Equation 5 aggregates *all* pairwise relations, so even a small region of mis-aligned samples suppresses the score which CKA can be too strict for measuring cross-modal cases.

### B.2.2 Centered-Kernel $k$-NN Alignment (CKNNA)

To focus on *local* structure, the authors in [10, 1] suggest a binary mask $\alpha(i, j)$ that selects only pairs that are *mutual* $k$-nearest neighbours in **both modalities**:

$$\alpha(i, j) = \mathbb{1}\big[v_j \in \text{kNN}(v_i) \ \wedge \ l_j \in \text{kNN}(l_i) \ \wedge \ i \neq j\big]. \tag{7}$$

Using this mask, one can restrict the definition of alignment to bias more towards the local structure through replacing the full HSIC trace by a *masked* cross-covariance.:

$$\text{Align}_{\text{local}}(K, L) = \sum_{i=1}^{n} \sum_{j=1}^{n} \alpha(i, j)\big(\langle v_i, v_j \rangle - \mathbb{E}_\ell \langle v_i, v_\ell \rangle\big)\big(\langle l_i, l_j \rangle - \mathbb{E}_\ell \langle l_i, l_\ell \rangle\big) \tag{8}$$

$$= \sum_{i=1}^{n} \sum_{j=1}^{n} \alpha(i, j)\, \bar{K}_{ij}\, \bar{L}_{ij}. \tag{9}$$

The final formulation of CKNNA becomes:

$$\text{CKNNA}(K, L) = \frac{\text{Align}_{\text{local}}(K, L)}{\sqrt{\text{Align}_{\text{local}}(K, K)\, \text{Align}_{\text{local}}(L, L)}}. \tag{10}$$

Note that CKNNA can be fully recovered to CKA for $k \to n$, since $\alpha(i, j) = 1$ for all $i \neq j$.

Thus, CKA asks: *"Are the two representations **globally** linearly related?"*; whereas CKNNA captures: *"Do the two models agree **locally** (i.e., on each point's neighbours)?"*.

### B.2.3 CCA (Canonical Correlation Analysis)

Let $X \in \mathbb{R}^{d_x \times n}$ and $Y \in \mathbb{R}^{d_y \times n}$ be column-wise zero-centered feature matrices produced by two representations (all means are removed so that covariance estimates are unbiased). Define sample covariance blocks:

$$\Sigma_{xx} = \tfrac{1}{n} X X^\top, \quad \Sigma_{yy} = \tfrac{1}{n} Y Y^\top, \quad \Sigma_{xy} = \tfrac{1}{n} X Y^\top. \tag{11}$$

CCA searches for weight vectors $\mathbf{w} \in \mathbb{R}^{d_x}$, $\mathbf{v} \in \mathbb{R}^{d_y}$ that maximise the *Pearson correlation* between projected variables:

$$\rho = \max_{\mathbf{w}, \mathbf{v}} \ \frac{\mathbf{w}^\top \Sigma_{xy} \mathbf{v}}{\sqrt{\mathbf{w}^\top \Sigma_{xx} \mathbf{w}} \sqrt{\mathbf{v}^\top \Sigma_{yy} \mathbf{v}}} \tag{12}$$

$$\text{s.t.} \quad \mathbf{w}^\top \Sigma_{xx} \mathbf{w} = 1, \ \mathbf{v}^\top \Sigma_{yy} \mathbf{v} = 1. \tag{13}$$

Using Lagrange multipliers we obtain the generalized eigenvalue problem:

$$\big(\Sigma_{xx}^{-1} \Sigma_{xy} \Sigma_{yy}^{-1} \Sigma_{yx}\big)\mathbf{w} \ = \ \rho^2\, \mathbf{w}, \quad \mathbf{v} = \rho^{-1} \Sigma_{yy}^{-1} \Sigma_{yx} \mathbf{w}, \tag{14}$$

whose top $k$ eigenvalues $\rho_1 \geq \rho_2 \geq \ldots \geq \rho_k$ are the **canonical correlations**. We report the maximum canonical correlation value (obtained from the first canonical component) as an alignment score.

**PCA-assisted CCA** Since CCA method suffers from high-dimensionality data structures [54], we first project each feature vectors onto its leading $r$ principal components:

$$\tilde{X} = P_x^\top X, \qquad \tilde{Y} = P_y^\top Y, \tag{15}$$

with $P_x \in \mathbb{R}^{d_x \times r}$ (orthonormal) obtained from linear PCA or kernel PCA using an RBF kernel $\kappa(\mathbf{z}, \mathbf{z}') = \exp[-\gamma \|\mathbf{z} - \mathbf{z}'\|^2]$. CCA is then run on $\tilde{X}, \tilde{Y}$, producing more stable correlations in high-dimensional regimes. In our experiments, we used $r = 50$ (i.e., feature dimension = 50), and $k = 50$ (i.e., number of canonical components = 50).

### B.2.4 SVCCA (Singular Vector Canonical Correlation Analysis)

SVCCA [8] replaces the heuristic PCA cut-off with a *data-dependent approach* via SVD:

1. Compute SVDs: $X = U_x \Sigma_x V_x^\top$, $Y = U_y \Sigma_y V_y^\top$.
2. Retain the top $r_x, r_y$ singular vectors explaining at least a fixed proportion $\eta$ (e.g., 99%) of variance:
$$\hat{X} = \Sigma_{x,[1:r_x]} V_{x,[1:r_x]}^\top, \qquad \hat{Y} = \Sigma_{y,[1:r_y]} V_{y,[1:r_y]}^\top.$$
3. Run linear CCA on $(\hat{X}, \hat{Y})$ to obtain canonical correlations $\{\rho_i\}_{i=1}^k$ with $k = \min(r_x, r_y)$.

The SVCCA score is reported using mean of these $k$ correlations, $\text{SVCCA} = \frac{1}{k}\sum_{i=1}^k \rho_i$, where we report SVCCA score with $k = 10$.

### B.3 Supplemental Results for Statistical Alignment Tests

We describe the specific data pair set-up and provide results of statistical alignment tests for each specific task (i.e., Replace, Add, and Swap cases) in Sugarcrepe data [4]. In our experiments, we reduced features dimension to 50 for CCA using linear and kernel PCA; for SVCCA, CKA, and CKNNA, the features dimension was set to 10, and nearest neighbors $k = 10$ (same setting utilized in [1]).

#### B.3.1 Replace Task

- Match: $V$ and $L$ that are matching (i.e., correct correspondence)
- Easy Non-Match: $V$ and $L$ that have clear non-matching aspects
  - Case 1: $V$ = White-noise image embeddings, $L$ = Same $L$ from Match case
  - Case 2: $V$ = Same $V$ from Match case, $L$ = Text embeddings extracted from "The Great Gatsby" novel
- Hard Non-Match: $V$ and $L$ that are non-matching due to specific *attribute, object, and relation* being *replaced* by incorrect text; hence the overarching context of this text is similar to correct matching text, but only differ by the specific aspect. Thus, more fine-grained understanding is required to discern that it is actually a non-matching text for the image.

Table 7: Replace Task: Statistical Alignment Test Results

| Model Pairs | Match | | | | Easy Non-Match | | | | Hard Non-Match | | | |
|---|---|---|---|---|---|---|---|---|---|---|---|---|
| | CCA (linear/kernel pca) | CKA | SVCCA | CKNNA | CCA | CKA | SVCCA | CKNNA | CCA | CKA | SVCCA | CKNNA |
| Gemma2 & DINOv2 | 0.94 / 0.94 | 0.286 | 0.432 | 0.209 | 0.48 / 0.47 | 0.020 | 0.100 | 0.006 | 0.94 / 0.94 | 0.293 | 0.403 | 0.194 |
| Llama3.2 & DINOv2 | 0.94 / 0.95 | 0.447 | 0.425 | 0.275 | 0.47 / 0.47 | 0.027 | 0.097 | 0.010 | 0.94 / 0.94 | 0.431 | 0.444 | 0.240 |
| OLMo2 & DINOv2 | 0.95 / 0.95 | 0.439 | 0.505 | 0.287 | 0.47 / 0.47 | 0.028 | 0.102 | 0.008 | 0.95 / 0.95 | 0.422 | 0.482 | 0.252 |
| Gemma2 & ResNet50 | 0.91 / 0.91 | 0.25 | 0.430 | 0.190 | 0.51 / 0.51 | 0.028 | 0.100 | 0.006 | 0.91 / 0.91 | 0.250 | 0.420 | 0.190 |

#### B.3.2 Add task

- Match: $V$ and $L$ that are matching (i.e., correct correspondence)
- Easy Non-Match: $V$ and $L$ that have clear non-matching aspects
  - Case 1: $V$ = White-noise image embeddings, $L$ = Same $L$ from Match case
  - Case 2: $V$ = Same $V$ from Match case, $L$ = Text embeddings extracted from "The Great Gatsby" novel
- Hard Non-Match: $V$ and $L$ that are non-matching due to specific *attribute, object* being *added*, which leads to incorrect correspondence of reality

Table 8: Add Task: Statistical Alignment Test Results

| Model Pairs | Match | | | | Easy Non-Match | | | | Hard Non-Match | | | |
|---|---|---|---|---|---|---|---|---|---|---|---|---|
| | CCA | CKA | SVCCA | CKNNA | CCA | CKA | SVCCA | CKNNA | CCA | CKA | SVCCA | CKNNA |
| Gemma2 & DINOv2 | 0.95 / 0.94 | 0.311 | 0.425 | 0.201 | 0.48 / 0.47 | 0.022 | 0.095 | 0.007 | 0.94 / 0.94 | 0.323 | 0.448 | 0.206 |
| Llama3.2 & DINOv2 | 0.95 / 0.95 | 0.451 | 0.424 | 0.257 | 0.48 / 0.48 | 0.027 | 0.097 | 0.011 | 0.95 / 0.94 | 0.444 | 0.447 | 0.259 |
| OLMo2 & DINOv2 | 0.94 / 0.94 | 0.448 | 0.491 | 0.269 | 0.47 / 0.47 | 0.028 | 0.097 | 0.009 | 0.94 / 0.94 | 0.448 | 0.481 | 0.265 |
| Gemma2 & ResNet50 | 0.89 / 0.89 | 0.309 | 0.420 | 0.199 | 0.51 / 0.51 | 0.028 | 0.100 | 0.006 | 0.89 / 0.89 | 0.320 | 0.400 | 0.200 |

### B.3.3 Swap task

- Match: $V$ and $L$ that are matching (i.e., correct correspondence)
- Easy Non-Match: $V$ and $L$ that have clear non-matching aspects
  - Case 1: $V$ = White-noise image embeddings, $L$ = Same $L$ from Match case
  - Case 2: $V$ = Same $V$ from Match case, $L$ = Text embeddings extracted from "The Great Gatsby" novel
- Hard Non-Match: $V$ and $L$ that are non-matching due to specific *attribute, object* being *swapped*

Table 9: Swap Task: Statistical Alignment Test Results

| Model Pairs | Match | | | | Easy Non-Match | | | | Hard Non-Match | | | |
|---|---|---|---|---|---|---|---|---|---|---|---|---|
| | CCA | CKA | SVCCA | CKNNA | CCA | CKA | SVCCA | CKNNA | CCA | CKA | SVCCA | CKNNA |
| Gemma2 & DINOv2 | 0.95 / 0.95 | 0.317 | 0.440 | 0.228 | 0.47 / 0.47 | 0.037 | 0.124 | 0.014 | 0.94 / 0.94 | 0.330 | 0.433 | 0.220 |
| Llama3.2 & DINOv2 | 0.94 / 0.94 | 0.463 | 0.473 | 0.311 | 0.47 / 0.47 | 0.049 | 0.129 | 0.012 | 0.94 / 0.94 | 0.453 | 0.430 | 0.287 |
| OLMo2 & DINOv2 | 0.95 / 0.94 | 0.466 | 0.462 | 0.312 | 0.480 / 0.48 | 0.05 | 0.130 | 0.012 | 0.94 / 0.94 | 0.455 | 0.511 | 0.276 |
| Gemma2 & ResNet50 | 0.89 / 0.89 | 0.315 | 0.430 | 0.221 | 0.50 / 0.50 | 0.029 | 0.123 | 0.014 | 0.89 / 0.89 | 0.315 | 0.428 | 0.220 |

## C  Spread Loss Formulation for Multimodal Framework

### C.1  Our Formulation of Spread Loss for Multimodal Representation Learning

$$\mathcal{L}_{spread} = \frac{1}{2}\left[\mathcal{L}_{spread-\text{VL}} + \mathcal{L}_{\text{LV}}\right] \tag{16}$$

The formulation for VL direction is a weighted sum with $\alpha$ controlling the trade-off between context-level alignment and intra-context contrast:

$$\mathcal{L}_{spread-\text{VL}} = (1-\alpha)\mathcal{L}_{ConCon} + \alpha\mathcal{L}_{contextNCE} \tag{17}$$

For $\alpha \in [0,1]$, per-sample case of $\mathcal{L}_{spread-\text{VL}}$ is defined as:

$$\mathcal{L}_{spread-\text{VL}}(\Phi_V, v_i, B) = (1-\alpha)\mathcal{L}_{ConCon}(\Phi_V, v_i, B) + \alpha\mathcal{L}_{contextNCE}(\Phi_V, v_i, B) \tag{18}$$

where

$$\mathcal{L}_{ConCon}(\Phi_V, v_i, B) = -\frac{1}{|C_L(i)|}\sum_{c_l \in C_L(i)} \log\left(\frac{\sigma(v_i, c_l)}{\sigma(v_i, c_l) + \sum_{\widetilde{c_l} \in \widetilde{C}_L(i)} \sigma(v_i, \widetilde{c_l})}\right), \tag{19}$$

$$\mathcal{L}_{contextNCE}(\Phi_V, v_i, B) = -\log\left(\frac{\sigma(v_i, l_{P_i})}{\sum_{c_l \in C_L(i)} \sigma(v_i, c_l)}\right). \tag{20}$$

The overall loss $\mathcal{L}_{spread-\text{VL}}(\Phi_V, B)$ is computed by averaging over all $(v_i, l_{P_i}, l_{N_i}) \in B$:

$$\mathcal{L}_{spread-\text{VL}}(\Phi_V, B) = \frac{1}{|B|}\sum_{i=1}^{|B|}\mathcal{L}_{spread-\text{VL}}(\Phi_V, v_i, B). $$

Recall that since hard-negative texts do not have correct corresponding images, the formulation for LV direction omits hard-negative texts. So, it follows the standard LV loss in contrastive learning scheme of CLIP models:

$$\mathcal{L}_{\text{LV}}(\Phi_L, B) = -\frac{1}{|B|}\sum_{i=1}^{|B|}\log\frac{\sigma(l_{P_i}, v_i)}{\sum_{j=1, j\neq i}^{N}\sigma(l_{P_i}, v_j)} \tag{21}$$

### C.2  Spread Loss Formulation in Visual Representation Learning Domain

Spread loss in vision representation learning domain [32] is constructed by a weighted sum of a supervised contrastive loss ($\mathcal{L}_{\text{sup}}$) [34] and a class-conditional InfoNCE loss ($\mathcal{L}_{\text{cNCE}}$) [17]. The core motivation is similar in our multimodal scheme: training encoder to produce representations of the data by pulling together similar points (positive pairs) and pushing apart dissimilar points (negative pairs).

Formally, let $B$ be a batch of data from dataset $\mathcal{D}$. Define the positive set:

$$P(i, B) = \{x^+ \in B \setminus \{x_i\} : h(x^+) = h(x_i)\}$$

and the negative set:

$$N(i, B) = \{x^- \in B \setminus \{x_i\} : h(x^-) \neq h(x_i)\},$$

where $h(x)$ denotes the class label of $x$ and $a(x_i)$ be an augmentation of $x_i$. Define the similarity (i.e., cosine similarity) with temperature hyperparameter $\tau > 0$:

$$\sigma_f(x, x') = \exp\left(\frac{f(x)^\top f(x')}{\tau}\right).$$

For $\alpha \in [0, 1]$, the per-sample spread loss is defined as:

$$\mathcal{L}_{spread}(f, x_i, B) = (1 - \alpha)\mathcal{L}_{sup}(f, x_i, B) + \alpha\mathcal{L}_{cNCE}(f, x_i, B), \tag{22}$$

where

$$\mathcal{L}_{sup}(f, x_i, B) = -\frac{1}{|P(i, B)|} \sum_{x^+ \in P(i,B)} \log\left(\frac{\sigma_f(x_i, x^+)}{\sigma_f(x_i, x^+) + \sum_{x^- \in N(i,B)} \sigma_f(x_i, x^-)}\right), \tag{23}$$

$$\mathcal{L}_{cNCE}(f, x_i, B) = -\log\left(\frac{\sigma_f(x_i, a(x_i))}{\sum_{x^+ \in P(i,B)} \sigma_f(x_i, x^+)}\right). \tag{24}$$

The overall loss $\mathcal{L}_{spread}(f, B)$ is computed by averaging over all $x_i \in B$:

$$\mathcal{L}_{spread}(f, B) = \frac{1}{|B|} \sum_{x_i \in B} \mathcal{L}_{spread}(f, x_i, B).$$

$\mathcal{L}_{sup}$ encourages intra-class clustering by pulling together same-class samples, while $\mathcal{L}_{cNCE}$ repels within-class samples except for augmentations. Their combination spreads same-class points while preserving instance-level attraction, promoting more structured representation spaces.

### C.3 Side-by-Side Comparison

**Intuition.** $\mathcal{L}_{ConCon}$ *pulls* an image toward all texts that share its visual context (analogous to class-level clustering, performed by $\mathcal{L}_{sup}$), while $\mathcal{L}_{contextNCE}$ *pushes* it away from other texts inside that context except its primary caption (analogous to instance discrimination, performed by $\mathcal{L}_{cNCE}$).

Table 10 summarizes the symbols used in the Spread loss in vision domain (left column) and their direct counterparts in our our multimodal formulation (right column). The first row clarifies that the visual setting relies on a single encoder $f(\cdot)$, whereas the multimodal variant distinguishes between an image encoder $\Phi_V$ and a text encoder $\Phi_L$. Subsequent rows pair up the anchor, positive, and negative sets, making it explicit that class labels in the visual domain translate to visual contexts (collections of captions). Finally, the similarity function retains the same softmax-temperature structure; only the argument types differ (image-image vs. image–text).

Table 11 presents the four loss terms from an objective-driven perspective. Each row identifies the loss and breaks down its optimization goal into four parts: the anchor being updated, the positive samples it should align with, the negatives it should separate from, and the broader motivation behind this push-pull dynamic. This format highlights a consistent analogy: $\mathcal{L}_{sup}$ clusters images by class, while our $\mathcal{L}_{ConCon}$ clusters images with all captions from the same visual context. In contrast, $\mathcal{L}_{cNCE}$ and $\mathcal{L}_{contextNCE}$ act as sharpening losses, refining clusters by contrasting a specific positive — an augmentation for vision case and a primary caption for our multimodal case — against hard negatives.

Table 10: Notation Comparison of Spread Loss in Vision vs. Multimodal Domain (Ours)

| Symbol | Vision Spread | Multimodal Spread (Ours) |
|---|---|---|
| Encoders | $f(\cdot)$ | $\Phi_V(\cdot),\ \Phi_L(\cdot)$ |
| Anchor | $x_i$ (i.e., image) | $v_i = \Phi_V(x_i)$ |
| Positives | $P(i)$ same class | $C_L(i) = \{l_{P_i}, l_{N_i}\}$ (i.e., similar context texts) |
| Negatives | $N(i)$ diff. class | $\widetilde{C}_L(i) = L \setminus C_L(i)$ |
| Similarity | $\sigma(x, x')$ | $\sigma(v, l)$ |

Table 11: "Objective" view of each loss term.

| Loss | Anchor | Positives | Negatives | Goal |
|---|---|---|---|---|
| $\mathcal{L}_{sup}$ | $x_i$ | same class | other classes | intra-class cohesion |
| $\mathcal{L}_{ConCon}$ | $v_i$ | $C_L(i)$ | $\widetilde{C}_L(i)$ | cross-modal cohesion |
| $\mathcal{L}_{cNCE}$ | $x_i$ | $a(x_i)$ | other $P(i)$ | instance sharpening |
| $\mathcal{L}_{contextNCE}$ | $v_i$ | $l_{P_i}$ | $C_L(i) \setminus \{l_{P_i}\}$ | context sharpening |

# D  Model Analysis & Training Configuration

## D.1  Pretrained Unimodal Backbones

| LM | | | |
|---|---|---|---|
| Model | Size | # Layers | Dim |
| Gemma2 | 2B | 26 | 2304 |
| | 9B | 42 | 3584 |
| Llama3.2 | 1B | 16 | 2048 |
| | 3B | 32 | 4096 |
| OLMo2 | 7B | 32 | 4096 |
| | 13B | 40 | 5120 |

| VM | | | |
|---|---|---|---|
| Model | Size | # Layers | Dim |
| DINOv2 (SSL) | 86M (ViT-B) | 12 | 768 |
| | 300M (ViT-L) | 12 | 1024 |
| MAE (SSL) | 86M | 12 | 768 |
| Swav (SSL) | 23M | 50 | 2048 |
| Swin (Sup) | 88M | 50 | 2048 |
| ViT (Sup) | 86M | 12 | 768 |
| ResNet50 (Sup) | 23M | 50 | 2048 |

Table 12: Configuration of pretrained unimodal models used in experiments. For MAE, we used ViT as backbone architecture. For Swav and Swin, we used RestNet50 as backbone architecture.

## D.2  JAM (Joint Autoencoder Modulator) Framework

We experiment JAM (Joint Autoencoder Modulator) framework with 3 hidden layers with dimension size 512, bottleneck/latent layer with dimension size 256, drop out ratio of 0.1, batch size of 32, and SwiGLU activation. With this architecture, Table 13 shows the model analysis of JAM attached to each pretrained unimodal backbones.

With the extracted unimodal features (language and vision, respectively) data, for each task setting, we use 70-15-15 train/validation/test splits. For each task, we train our Joint Autoencoder Modulator (JAM) with all the loss schemes ($\mathcal{L}_{spread}$, $\mathcal{L}_{con}$, $\mathcal{L}_{NegCon}$) for 100 epochs with a batch size of 32, using data seeds 5, 42, and 55. The reported scores are the average of recall scores across different seeds. Both autoencoders are optimized jointly using AdamW [39] with gradient clipping (1.0) and a cosine annealing scheduler. We initialize the logit scaling parameter in log-space as $log(1/0.07)$, following the common CLIP [17] initialization strategy. During training, the effective scale is recovered via exponentiation, allowing the model to start with sharper similarity distributions and learn an appropriate temperature dynamically. The reconstruction loss is weighted by a linearly

decaying factor $\lambda(t)$, decreasing from 1.0 to 0.1 over training epochs to gradually emphasize the alignment objective. Every five epochs, we compute image-to-text Recall@1 on the validation set, applying early stopping if no improvement is observed for five consecutive validations. We evaluate on two retrieval settings: (1) binary choice between the positive and its hard negative (standard evaluation scheme in fine-grained task setting [4, 31]), and (2) a 5-way choice including three additional distractors.

| Pretrained Backbone | JAM Parameters (M) | JAM FLOPs (G) |
|---|---|---|
| Gemma2 (2B) | 11.55 | 2.39 |
| Gemma2 (9B) | 16.20 | 3.31 |
| Llama3.2 (1B) | 10.55 | 2.06 |
| Llama3.2 (3B) | 18.31 | 3.81 |
| OLMo2 (7B) | 18.31 | 3.81 |
| OLMO2 (13B) | 21.82 | 4.58 |
| DINOv2 (ViT-B) | 7.26 | 1.40 |
| DINOv2 (ViT-L) | 8.90 | 1.72 |

Table 13: JAM framework analysis attached to pretrained backbones

## E  Further Results

| Language Backbone (Model Size) | Vision Backbone (Model Size) | Alignment Method for JAM | Replace Task Recall@1 (binary) | Replace Task Recall@1 (5-way) | Add Task Recall@1 (binary) | Add Task Recall@1 (5-way) | Swap Task Recall@1 (binary) | Swap Task Recall@1 (5-way) |
|---|---|---|---|---|---|---|---|---|
| Gemma2 (2B) | MAE (SSL) | Con | 58.55 | 50.33 | 56.37 | 50.46 | 63.73 | 49.29 |
| | | NegCon | 77.62 | 69.38 | 89.15 | 80.16 | 70.81 | 58.63 |
| | | Spread | 89.09 | 71.16 | 94.35 | 84.98 | 79.05 | 58.63 |
| Gemma2 (2B) | Swav (SSL) | Con | 68.08 | 62.34 | 65.25 | 54.41 | 62.60 | 59.21 |
| | | NegCon | 83.31 | 76.53 | 95.20 | 91.63 | 80.75 | 67.04 |
| | | Spread | 88.88 | 76.98 | 95.83 | 90.64 | 81.01 | 67.70 |
| Gemma2 (2B) | Swin (Sup) | Con | 64.94 | 58.17 | 56.87 | 50.66 | 62.91 | 59.52 |
| | | NegCon | 85.07 | 78.35 | 96.39 | 91.66 | 77.93 | 70.92 |
| | | Spread | 85.16 | 79.01 | 97.87 | 92.22 | 80.32 | 71.46 |
| Gemma2 (2B) | ViT (Sup) | Con | 60.78 | 54.74 | 60.79 | 55.05 | 59.37 | 52.28 |
| | | NegCon | 86.06 | 75.89 | 96.31 | 92.50 | 59.21 | 49.58 |
| | | Spread | 86.35 | 75.26 | 95.37 | 89.62 | 82.31 | 67.72 |
| Gemma2 (9B) | DINOv2 (ViT-L; 300M) | Con | 68.77 | 63.95 | 59.31 | 54.58 | 60.92 | 51.85 |
| | | NegCon | 87.52 | 84.33 | 98.89 | 93.14 | 84.44 | 69.69 |
| | | Spread | 89.66 | 83.34 | 98.89 | 96.30 | 84.44 | 73.38 |
| Llama3.2 (1B) | MAE (SSL) | Con | 67.82 | 54.48 | 61.08 | 55.33 | 61.19 | 54.68 |
| | | NegCon | 82.92 | 74.09 | 94.16 | 84.80 | 76.77 | 68.26 |
| | | Spread | 86.23 | 74.96 | 96.85 | 92.03 | 80.47 | 68.67 |
| Llama3.2 (1B) | Swav (SSL) | Con | 65.94 | 58.97 | 68.12 | 62.93 | 75.39 | 68.03 |
| | | NegCon | 92.37 | 83.91 | 92.12 | 87.48 | 78.62 | 71.84 |
| | | Spread | 88.63 | 79.24 | 98.94 | 94.75 | 83.29 | 69.99 |
| Llama3.2 (1B) | Swin (Sup) | Con | 70.59 | 63.60 | 69.70 | 63.48 | 75.39 | 68.03 |
| | | NegCon | 86.01 | 83.05 | 93.89 | 88.39 | 69.11 | 63.45 |
| | | Spread | 87.24 | 76.69 | 94.72 | 94.16 | 77.62 | 68.84 |
| Llama3.2 (1B) | ViT (Sup) | Con | 66.81 | 57.49 | 64.50 | 60.70 | 64.87 | 59.79 |
| | | NegCon | 82.81 | 77.15 | 93.16 | 90.57 | 74.50 | 63.18 |
| | | Spread | 87.56 | 76.13 | 97.38 | 92.23 | 74.23 | 57.79 |
| Llama3.2 (3B) | DINOv2 (ViT-L) | Con | 65.24 | 59.14 | 62.38 | 57.9 | 73.12 | 65.18 |
| | | NegCon | 84.54 | 82.66 | 92.61 | 88.16 | 75.93 | 70.27 |
| | | Spread | 89.43 | 82.48 | 95.74 | 91.25 | 82.44 | 76.77 |
| OLMo2 (7B) | MAE (SSL) | Con | 66.88 | 57.37 | 68.12 | 56.98 | 58.36 | 51.85 |
| | | NegCon | 86.39 | 74.57 | 93.05 | 88.33 | 71.94 | 60.62 |
| | | Spread | 83.05 | 77.84 | 96.39 | 90.64 | 79.32 | 64.52 |
| OLMo2 (7B) | Swav (SSL) | Con | 66.38 | 61.12 | 67.38 | 62.58 | 63.75 | 58.65 |
| | | NegCon | 88.18 | 83.31 | 92.12 | 88.50 | 82.23 | 69.98 |
| | | Spread | 93.27 | 85.63 | 98.43 | 93.70 | 81.01 | 72.81 |
| OLMo2 (7B) | Swin (Sup) | Con | 65.35 | 59.45 | 70.82 | 63.79 | 55.67 | 50.85 |
| | | NegCon | 89.40 | 86.70 | 95.28 | 89.35 | 73.66 | 71.96 |
| | | Spread | 91.32 | 82.11 | 97.87 | 94.72 | 80.16 | 70.81 |
| OLMo2 (7B) | ViT (Sup) | Con | 70.56 | 64.79 | 69.98 | 58.86 | 69.15 | 60.36 |
| | | NegCon | 84.61 | 80.77 | 94.16 | 87.86 | 81.74 | 67.30 |
| | | Spread | 89.25 | 81.01 | 98.43 | 95.18 | 76.49 | 66.44 |
| OLMo2 (13B) | DINOv2 (ViT-L) | Con | 65.86 | 68.68 | 59.21 | 62.52 | 62.28 | 52.12 |
| | | NegCon | 89.36 | 82.68 | 93.7 | 91.66 | 78.89 | 67.57 |
| | | Spread | 90.53 | 83.18 | 97.96 | 92.42 | 84.71 | 69.69 |

Table 14: Image-to-Text Retrieval Results of Joint Autoencoder Modulator (JAM) for Vision-Language Compositionality with wider set of pretrained backbones.

## F  Compute Systems/Resources

All experiments were conducted in either set-ups: Apple Macbook Pro (M2 chip) or NVIDIA RTX 3080. We utilized NVIDIA RTX 3080 system for language models' feature extraction and M2 chip

system for vision models' feature extraction. Running our JAM framework is done in both set-ups. Finetuning CLIP model is conducted in NVIDIA RTX 3080 system.

