# OpenReview forum: "Escaping Plato’s Cave: JAM for Aligning Independently Trained Vision and Language Models"
_NeurIPS.cc/2025/Workshop/UniReps — UniReps2025_

### Official Review · Reviewer_TtNE · 2025-09-15
**An interesting and topically relevant paper that would benefit from some clarification**

**Confidence:** 3

**Review:**

The authors highlight a problem wtih standard representational alignment across unimodal vision and language models: that the alignment does not reflect fine-grain distinctions across modality. They then introduce a new architecture and training loss to align unimodal vision and language models with minimal image-caption pairs.

The issues with representational alignment and proposed solutions are all very interesting and relevant to the workshop, and the technical contributions seem solid. The paper could be written more clearly, both in terms of stressing the gap it addresses and solution, as well as explaining the technical details. This lack of clarity somewhat limits its impact. Specific comments below:

- The abstract and intro both state that vision/lang models "typically inhabit disjoint representational spaces" but it was not clear waht this meant (particularly in the context of platonic rep hypothesis). Are you referring to input representations in particular?

- Lines 44-47. This wording is unclear / inaccurate. Unimodal models do capture discriminative fine-grain details (within each modality). They just don't do it in a way that aligns across vision/lang. This lack of clarity early on made it very hard to understand the (important) gap in the introduction

- Lines 91-92. It was not clear why investigating alignment post-hoc (without multimodal inductive biases) was important. Later the authors mention that this alignment can be achieved with relatively little training data, compared to multimodal training in CLIP-style models (lines 271-273) Perhaps this is related and if so it should be mentioned earlier?

- Loss function description: The terms L_VL and L_LV seem to be used interchangeably, which was confusing. While Figure 4 is a helpful overview it was not entirely clear how the initial LV (or VL?) loss maps onto this. A better intuitive understanding of Loss_Spread and Loss_spread-VL would help. Also InfoNCE-style objective is mentioned but it does not seem like it is ever defined or that a reference is provided.

- Figure 2: It was not clear why the circles for different models were different sizes. To be avoid obstruction if they overlapped? Or did the size denote something special about each model? It would also help to label the y-axis "alignment with vision model embeddings"

**Score:**

4

**Topic Fit:**

3

---

### Official Review · Reviewer_erXu · 2025-09-15
**Very interesting alignment method with uncommon evaluation setting and some unconnected experiments**

**Confidence:** 4

**Review:**

**Summary:**

The paper introduces a model that combines a pre-trained unimodal vision model and language model for fine-grained compositional tasks. It shows that hard negatives (text captions where some parts of the caption are altered) have similar representational alignment scores as the ground truth captions. Furthermore, it proposes a new alignment module which is trained on a combination of a reconstruction loss with a separate contrastive loss for hard negatives and easy negative. The method is evaluated on the SugarCrepe and Winoground datasets and shows that hard negatives are better discriminated than with CLIP. Finally, an ablation study on the loss, the mixing hyperparameter in the loss and the size of the backbone networks is performed.

**Strengths:**

1. The Spread loss separates the easy negatives from the hard negatives. This is a useful extension of the NegCon loss from NegCLIP.
2. Section 3 shows that with some alignment metrics, the alignment for hard negatives is similar as for the ground-truth captions.
3. The model performs better than different CLIP variants on their SugarCrepe and Winoground setting.
4. Section 3 introduces all relevant loss functions and is easy to understand.
5. Figure 1, Figure 3 and Figure 4 a good overviews of the task, method, and loss, respectively.
6. The method doesn’t require a lot of compute. It can be trained on a single NVIDIA RTX 3080 or on a Macbook Pro (M2).

**Major Weaknesses:**

1. The experimental setting are not consistent and is not compared with previous work except CLIP and NegCLIP. The methodological contribution of the paper is two-fold: (1) It combines two unimodal models instead of a CLIP model and (2) it introduces the Spread loss. For (1), there a numerous methods that archive CLIP-like capabilities with unimodal vision and language encoders, e.g. BLIP/BLIP-2 or ASIF [1]. Therefore, it would interesting if these methods also have the same problems as CLIP with fine-grained compositional tasks. For (2), it makes sense to compare to newer loss functions for compositional tasks like CE-CLIP [2] and TripletCLIP [3]. However, to my understanding, these methods usually train on automatic variations of other paired datasets like COCO without fine-tuning on the SugarCrepe dataset. This marks a deviation from this line of work and is therefore not comparable. Finally, these methods should also be introduced in the related work section.
2. Section 2 shows that the alignment for hard negatives is similar to the alignment of ground-truth text embeddings. However, the paper also notes that this is “not a failure of alignment but a reflection of the metrics themselves: hard negatives share global semantics with positives, differing only in fine-grained attributes.” (ll. 117-119). So, the alignment measures focus too much on the global alignment instead of the local alignment. This argument could be explained in more detail in particular for CKNNA, which is a local metric. Also, if this is true, the experiments don’t show anything about the alignment of discriminative details. While it seems to be true that fine-grained structures are not always embedded similarly in unimodal vision and language models (e.g. [4]), some more empirical evidence would be helpful.
3. In general, the connection of Section 2 and the Platonic representation hypothesis (PRH) to the JAM method is nor clear. The PRH is introduced as “a shared semantic structure that exists despite the models being trained in isolation.” (ll. 41-43). However, the JAM method is trained on paired data, even utilizing the knowledge about hard negatives. Also, it is not evaluated whether JAM improves the alignment in the experiment of Section 2. Therefore, these sections are quite isolated and the paper doesn’t build a good connection between Section 2 (and the PRH) and the remaining paper.
4. The claim that “$L_{spread}$ enables learning representations that are not only globally aligned across modalities but also sensitive to subtle mismatches in local content” (ll. 230-231) is not fully backed by the experiments. It is shown that fine-grained retrieval gets better but not that the embeddings are globally aligned.
5. In the abstract it is claimed that “theoretical insight into the structure of shared semantics” (ll. 16-17) will be provided. However, this is quite vague and it is not clear what experiment gives more insights on that.

**Minor Weaknesses:**

1. Section 2 is hard to read. In lines 118-112, a lot of notation is introduced, which is only defined in Table 1, which appears one page later. It would be easier to read if every symbol would be introduced in textual form at this position, e.g.: “We construct the dataset $D$ in the nested pair format: $D = \dots$, where $v_i$ is the image embedding, $l_{P_i}$ is the positive text embedding, and $l_{N_i}$ is its hard negative text embedding. We denote the set of similar text embeddings as $C(i) = \{l_{P_i}, l_{N_i}\}$ and the set of other dissimilar text embeddings as $\widetilde{C}(i) = L \setminus C(i)$, where $L$ is the set of all text embeddings”. The comment about the experiments in Table 2 and Table 7, 8, 9 in the Appendix in ll. 111-112 also doesn’t fit well in the context because the experiment was described in the paragraph before.
2. The notation is not always consistent, e.g. $C_L(i)$ is introduced in Section 2 as $C(i)$) and overly complex ( $\Phi_V, \Psi_V, \Phi_L, \Psi_L, V, L_P, L_N$ are not used in the main paper).
3. Concerning Section 4.3: Would it make sense to evaluate different sizes of the same model family, e.g. different sizes of the LLaMA 3.2 models?
4. Many methods also evaluate on the newer variation SugarCrepe++ [5], which also adds an additional positive text sample which is lexically different. It seems like some methods perform good on SugarCrepe even though the performance drops on SugarCrepe++.

**Justification For Recommendation:**

The rating of weak reject is mainly based on missing evaluations on standard evaluation settings and the incoherency of Section 2 and Section 3.

**References:**

[1] Norelli, Antonio, et al. "Asif: Coupled data turns unimodal models to multimodal without training." Advances in Neural Information Processing Systems 36 (2023): 15303-15319.

[2] Zhang, Le, Rabiul Awal, and Aishwarya Agrawal. "Contrasting intra-modal and ranking cross-modal hard negatives to enhance visio-linguistic compositional understanding." Proceedings of the IEEE/CVF Conference on Computer Vision and Pattern Recognition. 2024.

[3] Patel, Maitreya, et al. "Tripletclip: Improving compositional reasoning of clip via synthetic vision-language negatives." Advances in neural information processing systems 37 (2024): 32731-32760.

[4] Schnaus, Dominik, Nikita Araslanov, and Daniel Cremers. "It's a (Blind) Match! Towards Vision-Language Correspondence without Parallel Data." Proceedings of the Computer Vision and Pattern Recognition Conference. 2025.

[5] Dumpala, Sri Harsha, et al. "Sugarcrepe++ dataset: Vision-language model sensitivity to semantic and lexical alterations." Advances in Neural Information Processing Systems 37 (2024): 17972-18018.

**Score:**

2

**Topic Fit:**

2

---

### Official Review · Reviewer_6J8G · 2025-09-16
**Provides a practical and promising framework for fine-grained multi-modal alignment research**

**Confidence:** 3

**Review:**

- **Summary:** The authors point out that research has been focused on statistical metrics that can detect alignments but are not sensitive enough to capture the discriminative detail. They propose a framework using autoencoders called JAM that aligns frozen, independently trained vision and language models. JAM couples modality-specific reconstruction with a shared latent space and introduces a multimodal Spread Loss to handle fine-grained distinctions. JAM improves fine-grained image–text retrieval and rivals/surpasses several CLIP variants; ablations highlight the role of reconstruction and alignment objective over the model scale.
- **Originality:** Though elements of the framework (e.g. autoencoders and supervised contrastive learning objectives) are mostly drawn from known techniques, the post-hoc, joint-autoencoder alignment of unimodal backbones and the adaptation of spread-style supervision to multimodal learning are novel.
- **Clarity:** The paper is generally clearly written. However, directly mentioning that a hard non-match is the hard negative in line 97 (as has been done with the match — true positive on line 96) will make it more consistent. I understand the small font sizes for the figure captions due to the page limit, but I find label texts in figures (e.g. Fig 2a to 2d, Fig 5, Fig 6) are very small as well. Making them bigger would make the figures easier to read.
- **Technical soundness:** The approach seems technically solid and well-controlled. The partial asymmetry in the Spread Loss (VL emphasis) and retrieval-only evaluation are reasonable but limit the scope.
- **Quality of results:** Main findings are solid. Curriculum experiments and model-scale probes add insight, though error bars and broader benchmarks would strengthen claims. I would love to see a discussion section, but considering this is a workshop and there is a page limit, I am satisfied with what the authors have done.
- **Significance:** Offers a practical method to turn strong unimodal foundation models into competent multi-modal models without costly end-to-end pretraining.

**Score:**

4

**Topic Fit:**

3

---

### Official Review · Reviewer_HiZM · 2025-09-16
**Aligning Independently Trained Vision and Language Models**

**Confidence:** 4

**Review:**

# Summary

The paper introduces the Joint Autoencoder Modulator (JAM), a post-hoc framework for aligning independently trained vision and language models. Inspired by the Platonic Representation Hypothesis, which posits that unimodal models may converge toward a shared statistical model of reality, the authors propose explicitly optimizing alignment rather than only diagnosing it. JAM leverages modality-specific autoencoders, reconstruction objectives, and a novel Spread Loss to capture both coarse and fine-grained semantic distinctions. Empirical results on SugarCrepe and Winoground benchmarks show JAM consistently outperforms contrastive baselines and even rivals CLIP variants in fine-grained compositional reasoning tasks.

# Strengths

- Methodological Innovation: JAM is lightweight, requiring no end-to-end multimodal pretraining. Spread Loss effectively balances global semantic alignment with fine-grained contrast, outperforming contrastive and NegCon baselines.
- Practical Implications: JAM offers a path for reusing frozen unimodal foundations, which is especially relevant in low-resource or specialist domains where paired multimodal data is scarce.

# Limitations

- Benchmark Coverage: While SugarCrepe and Winoground are appropriate for compositionality, reliance on only two datasets may limit generalizability. Evaluation on larger or more diverse multimodal benchmarks (e.g., Flickr30k, COCO retrieval, SugarCrepe++, VG Relations/Attributions, etc...) would strengthen claims.
- Comparisons to Modern Alignment Methods: The work compares mostly to CLIP and fine-tuning variants. It would be useful to also compare against recent adapter-based multimodal alignment methods or post-hoc alignment works beyond NegCLIP.
- Limited Benchmarks Tasks: The authors do not evaluate on coarse benchmarks like zero-shot object recognition (e.g., ImageNet). On such tasks, where unimodal backbones already excel, JAM offer any benefits in-regard to these tasks, or does JAM come with a cost of degrading performance on other tasks.

**Score:**

3

**Topic Fit:**

3